# Optical tissue clearing and 3D imaging of intact primate testicular tissue: a novel technology development

Pauline Wanjiku Kibui[1], Sarah Weischer[2], Nicola von Ostau[3], Jochen Hess[3], Nils Kirschnick[4], Thomas Zobel[2], Stefan Schlatt[1]*

1 Centre of Reproductive Medicine and Andrology, University of Münster, Münster, North Rhine-Westphalia, Germany, 2 Münster Imaging Network, Cells in Motion Interfaculty Centre, University of Münster, Münster, North Rhine-Westphalia, Germany, 3 University Clinic Essen, Clinic and Policlinic for Urology, Children Urology and Urooncology, Essen, Germany, 4 BioOpticService Unit, Max Planck Institute for Molecular Biomedicine, Münster, North Rhine-Westphalia, Germany

* Stefan.Schlatt@ukmuenster.de

## Abstract

Classical histology struggles to preserve three-dimensional spatial context, prompting the emergence of optical tissue clearing techniques that enable imaging of intact specimens at cellular or subcellular resolution. These techniques have revolutionised fields like cell biology, developmental biology, and neuroscience. However, their application in reproductive biology remains unexplored – particularly in studying the complexities of testicular development. We developed a novel, efficient and affordable toolbox for studying intact testicular tissues, PT-CLEAR3D, that stands for primate testis – whole mount staining, tissue clearing and three-dimensional imaging. Intact testicular tissues from humans (transgender model), common marmosets and macaques underwent antibody labelling, clearing with organic solvents, and three-dimensional imaging using light sheet fluorescence microscopy. Marker specificity was confirmed by immunofluorescence staining of 3 and 25 μm testicular sections, followed by imaging with confocal microscopy. The testicular structure was evaluated using several markers: spermatogonia (melanoma-associated antigen 4), least differentiated spermatogonia (Piwi-like protein 4), Sertoli cells (vimentin and SRY-Box transcription Factor 9), peritubular myoid cells and vasculature (alpha-smooth muscle actin), and NucSpot as a nuclear dye. PT-CLEAR3D efficiently achieved optical transparency while a commercial kit that was run in parallel was inefficient. This study presents a pioneering three-dimensional visualization of intact testicular samples of up to 50 mm³ in size and imaging depth of up to 4.5 mm across three primate species. Remarkably, PT-CLEAR3D revealed critical details at both tissue and cellular levels such as the spatial distribution of germ and somatic cells, cellular bridges, and vasculature. Furthermore, PT-CLEAR3D enabled three-dimensional reconstructions that effectively reduces confirmation bias enhancing our observation of spermatogonial

**Data availability statement:** All relevant data are within the paper and its Supporting Information files.

**Funding:** The Deutsche Forschungsgemeinschaft (DFG) provided funding for staff and consumables (Clinical Research Unit #326) and for instrumentation: Zeiss LSM 980: INST 211/898-1 FUGB (Confocal Laser Scanning Microscope) and Miltenyi UltraMicroscope II Super Plan: INST 211/899-1 FUGB (Light Sheet Microscope). We acknowledge support from the open access publication fund of the University of Münster.

**Competing interests:** No competing interests.

clones organized as single cells, pairs, and quartets. Importantly, it adeptly identified testicular pathology and the persistence of germ cell clones in select tubules within the transgender testis following hormonal suppression of spermatogenesis. This technological development offers a versatile toolbox with benefits such as applicability across multiple species, fluorophore multiplexing, compatibility with different fixatives and deep tissue volumetric imaging with cellular resolution. Overall, PT-CLEAR3D establishes a foundation for spatial evaluation of testicular development, presenting substantial potential for advancing our understanding of the intricate kinetics of spermatogenesis in health and disease.

## Introduction

Classical histology has been crucial for both basic and clinical science, enabling the examination of tissue structure, function, and disease diagnosis for over two decades [1,2]. This method involves manual sectioning of ultra-thin 2D slices from opaque biological specimen [3]. A key limitation of classical histology is its inability to preserve the three-dimensional (3D) spatial context of tissues, leading to the development of optical tissue clearing techniques [4]. Moreover, light scattering limits the imaging depth of fluorescence microscopy and often precludes imaging of intact whole mount tissue using confocal or 2-photon microscopy. Optical clearing techniques have intensified the desirability to image large specimens at cellular or subcellular resolution while preserving tissue integrity [3]. The overall aim of such techniques is to minimize light scattering and absorption by minimizing refractive index (RI) differences between components within biological samples [5,6]. Recent advancements in optical clearing, coupled with high-resolution imaging platforms like light sheet fluorescence microscopy (LSFM), have greatly impacted fields such as cell biology and neuroscience enabling computational data visualization and analysis [2,5,7]. However, their application in reproductive biology, particularly in the complex primate testis, remains largely unexplored.

The testis is a bicompartmental organ [8], which makes visualizing both the tubular compartment housing various germ cell types and the interstitial compartment containing different somatic cells difficult. The high cellular density in the testis affects opacity due to high cellular components such as lipids that cause light scattering. A delicate interaction between germ and somatic cells is essential for spermatogonial proliferation, differentiation, and renewal [9,10]. While there is significant information on spermatogenesis and testicular organization in primates, much research relies on microtome tubular cross-sections and small tissue samples [11,12]. Although tubular cross-sections offer versatility and ease of use, they have significant drawbacks such as tissue distortion, mixed cell types that complicate analysis, and missing generations of germ cells [9]. In contrast, whole mount preparations provide better insights into nuclear morphology and the topographical arrangement of spermatogonia compared to tubular cross-sections [13]. However, both microtome sections and whole mount preparations fail to accurately represent cellular density and associations [12].

Additionally, attempts to circumvent the challenge of tissue opacity through 3D reconstruction of testicular tissues from serial sections are flawed [10].

Previous attempts to address the above challenges in studying spermatogonial population kinetics in the primate testis included the use of camera lucida drawings [10]. In 1911, the Spalteholz clearing technique was introduced that involved bleaching tissues with 10% hydrogen peroxide followed by clearing using a mixture of benzyl benzoate and methyl salicylate [14]. Although this technique was successful, it caused significant damage to the tissues and resulted in blurring, which hindered further histological investigations. In 2004, Pais and colleagues used a modified Spalteholz technique to clear human testis samples but their mechanical sectioning approach faced the limitations of classical histology and only allowed for the visualization of vasculature from 2D serial sections [15]. Recently, new clearing techniques have been applied to intact rodent testes [16,17], but no data exists on the architectural organization of the testis using these methods. Currently, researchers primarily use tubular cross-sections and whole mount preparations of teased tubules to study primate testes. Despite significant progress in testis biology over the past fifty years, the mechanisms of spermatogenesis remain poorly understood. Therefore, developing advanced analytical techniques for studying primate testes in 3D, alongside traditional methods, could significantly improve our understanding of testicular development and the dynamics of the spermatogenic epithelium.

This study presents PT-CLEAR3D, an efficient and cost-effective protocol for the 3D visualization and analysis of intact testicular tissues (ITTs) from humans, common marmosets, and macaques. We used testicular tissue from transgender individuals undergoing gender-affirming hormonal treatment (GAHT) due to the limited availability of human testes for biomedical research, especially normal samples. We also included common marmosets and macaques, which are suitable preclinical models for studying human spermatogenesis, as they have been used in our previous studies [18,19]. The primary objective was to develop a user-friendly toolbox for imaging thick ITTs to enable visualization of the primate testes' tubular and interstitial compartments. By combining whole mount immunolabelling with optical tissue clearing and advanced imaging platforms, we successfully visualized the spatial organization of primate testes and their vasculature in both healthy and pathological states across three species.

## Materials and methods

### Tissue processing and ethical approval

For decades, CeRA maintains a monkey core facility, providing access to breeding groups of marmosets and macaques. The facility is licensed by the City of Münster under a permanent permit (53.5.32.7.1/MS-01433, granted: 8.11.2018; modified: 26.06.2019). The breeding core is linked to a non-human primate tissue bank that stores monkey tissue obtained from the core facility and samples from licensed studies granted by LANUV NRW. Human testicular tissue was obtained from two patients with gender dysphoria undergoing sex reassignment surgery from the urology department of the University Clinic Essen. Ethical approval was received from the ethics committee of the Ärztekammer Westfalen-Lippe (no. 2012–555-f-S). Written informed consent was obtained from both patients before study participation. All testicular tissues from humans, marmosets and macaques had been fixed either in Bouin's or paraformaldehyde (PFA) and were preserved in the tissue bank (70% ethanol vol/vol concentration). Fixed specimens were dissected into fragments ranging from 0.5 to 50 mm³ in size for whole mount staining and subsequent tissue clearing. Some fragments were paraffin embedded and cut into 3 and 25 μm thick microtome sections.

### Immunofluorescence of testicular sections

Immunofluorescence (IF) analysis of testicular sections from marmosets, macaques and humans was performed for two reasons. The first was to characterize tissue integrity as regards different germ and somatic cell populations and the second to ascertain marker specificity. Dewaxed sections were rehydrated in graded alcohol series and washed with

Tris-buffered saline (TBS) and distilled water. Antigen retrieval was performed in citrate buffer (pH 6) in the microwave oven for 15 minutes followed by cooling to room temperature. After washing in TBS, reactive aldehydes were blocked followed by permeabilization of the sections using 0.5% Triton at room temperature. Blocking buffer (5% donkey serum, 5% BSA, and 0.1% Tween in TBS) was used to block non-specific binding sites for 30 minutes at room temperature. Double immunofluorescence was done by incubating the sections overnight at 4°C in different primary antibodies. Markers include: testicular germ cell marker Melanoma Antigen family A 4, MAGEA4 (a gift from Prof. G.C. Spagnoli from the University Hospital of Basel Switzerland; dilution 1:20); peritubular myoid cell marker, mouse monoclonal anti-α-smooth muscle actin-Cy3 (α-SMA-Cy3) (C6198; Sigma-Aldrich; dilution: 1:200); Sertoli cell markers SRY-BOX9, SOX9 (AB5535; Merck; dilution 1:200) and anti-Vimentin VHH-Atto488 (vba488: Proteintech: dilution: 1:200). Negative control sections were treated with rabbit IgG (I5006; Sigma; dilution 1:1000) and mouse IgG (I5381; Sigma: dilution 1:1000) antibodies. Thereafter, secondary antibodies conjugated to fluorochromes (715546150; JacksonImmunoResearch; dilution; 1:100 and A31573; Invitrogen; dilution 1:100) were applied for 1 hour. Next, sections were washed and mounted with Vectashield containing 1.5 µg/mL diamidino-2-phenylindole (DAPI).

**Whole mount staining of intact testicular tissues**

Bouin's fixed marmoset ITTs of about 0.5 to 1 mm³ in size were stained with 5-bromo-2'-deoxyuridine (BrdU) as described by Ehmcke and Schlatt [18]. The BrdU incorporates into the nuclear DNA offering a valuable tool for the assessment of proliferating germ cells in the S-phase. As these tissues were not optically cleared, immunolabelling and imaging were inefficient prompting major modifications of the whole mount protocol followed by optical tissue clearing as described below.

For tissue clearing, Bouin's and PFA fixed ITTs ranging from 0.5 to 50 mm³ in size from adult humans (transgender model) and normal breeding marmosets and macaques were used. The intact testicular tissues were dehydrated in a graded series of alcohol: 1 hour in 85% ethanol, 1 hour in 95% ethanol, overnight in 100% ethanol, then 1 hour in 95%, 1 hour in 85%, and finally overnight in 70% ethanol. Dehydration helps to dissolve membranes for better staining diffusion. After, ITTs were incubated in hydrochloric acid (HCl) for 15 minutes and washed in distilled water. Next, they were treated with a trypsinizing solution for 30 minutes at room temperature, followed by another wash in distilled water and Tris-buffered saline (TBS) with 0.1% Triton X-100 for six hours with gentle shaking at room temperature. Then, ITTs were blocked for unspecific binding by incubation in 0.1% bovine serum albumin (BSA), 5% donkey serum and 0.5% Triton X-100 in TBS for two days. In whole mount staining, full sample penetration by the staining reagent is crucial. Thus, overnight blocking was done at 4°C and subsequent incubation was performed during the day at room temperature with gentle shaking as well as all the washing steps. After blocking, incubation with primary antibodies was done for two days—MAGEA4 or PIWIL4 (LS-C482396, LSBio, dilution: 1:100) followed by application of secondary antibodies, Donkey anti-mouse Alexa fluorochrome 647 (A31571, Invitrogen, dilution: 1:100) and/or Vimentin-Label Atto488 (vba488, Proteintech, dilution: 1:200) for two days. Following a six hour wash in TBS with 0.1% Triton X-100, samples were incubated with mouse anti-smooth muscle antibody Cy3 (C6198, Sigma-Aldrich, dilution: 1:200) and NucSpot 750/780 nuclear dye (41038-T, Biotium, dilution: 1:1000) for two days followed by a six hour wash in TBS with 0.1% Triton X-100. For this protocol, TBS was sterilized, and 0.1% Triton X-100 was added in the wash buffer throughout the protocol.

Our whole mount staining protocol, PT-CLEAR3D, was run in parallel with a commercial staining and clearing kit whose identity will be kept anonymous. For this comparison, Bouin's fixed ITTs were used. The same germ and somatic cell markers were used as described above. However, the kit provided all the buffers and most reagents. A few differences from our protocol include use of phosphate buffered saline (PBS) as the wash buffer, permeabilization of samples prior to dehydration and use of methanol for dehydration. For this, methanol dehydration series were 50% methanol in PBS, 80% methanol in deionized water and 100% in dry methanol.

 

Testicular samples with varying thickness and dimensions, width (X) x height (Y) x depth (Z) respectively, were subjected to these protocols. Notably, only the region of interest (ROI) is shown in these images and therefore, they do not represent the entire optically cleared tissue volumes. Human testicular tissues measuring 3 mm x 3 mm x 4 mm totaling 36 mm$^3$; 1 mm x 1 mm x 0.6 mm, totaling 0.6 mm$^3$; and 1 mm x 1 mm x 1 mm, totaling 1 mm$^3$ were used. The marmoset tissue had a volume of 29 mm$^3$ with 3.2 mm x 3.2 mm x 2.8 mm dimensions while the macaque tissue dimensions were 3.2 mm x 3.2 mm x 2.4 mm totaling 25 mm$^3$. The images shown in 3D, are either maximum intensity or volumetric projections of the tissues while single optical planes were used for the 2D, therefore, no projections were used.

## Optical tissue clearing of the primate testes

Finally, the last step was to homogenize the optical properties of the samples and their surrounding medium to minimize light scattering resulting from cellular and extracellular structures. Therefore, immunostained ITT samples were embedded in 1% low melting agarose (10-36-1010, Bio-Budget Technologies, Concentration: ≥ 200g/cm$^2$ gel starch) in a 24 well culture plate to support the fragile testicular tissues and cured. Rectangular blocks were cut around the samples and transferred to glass vials for clearing as described by Kirschnick and colleagues [5] with modifications. Briefly, dehydration was performed using a graded series of methanol (AE71, Carl Roth, Concentration: 99,95%): 50%, 70% in distilled water (vol/vol) for one hour each, followed by anhydrous 99.95% methanol, with gentle shaking overnight. Next, the samples were incubated in a 1:1 mixture of methanol and Benzyl Alcohol/Benzyl Benzoate (BABB) (108006, Sigma-Aldrich, Concentration: ≥ 99%/ B6630, Sigma-Aldrich, Concentration: ≥ 99.0% respectively), for two hours. Finally, the samples were incubated in 100% BABB that has a refractive index of 1.56 for four hours and stored in the dark. Optical clearing with the commercial kit involved incubating the immunostained ITTs in reagent 1 followed by reagent 2 for 12 hours each.

Samples were put on a microscope slide and placed on a grid to assess clearing efficiency before the clearing commenced and after. Clearing was considered efficient if the grids below the ITTs were visible through the tissues and inefficient if the grids were not visible. Images were taken before and after the optical tissue clearing using a cellular phone camera under a fume hood. It is noteworthy to mention that several (2–4) testicular tissues were processed for each species for each experimental run. Therefore, the tissue images shown may not represent the same testicular tissue before and after clearing since not all ITTs were photographed at each stage to minimize exposure of the immunostained ITTs to light.

## Fluorescent imaging with scanning microscopes

Fluorescence imaging of immunostained testicular sections (Fig 1) was performed using an Olympus VS120 S6 Slide scanner (Software: VS-ASW-S6) and Zeiss LSM 980 (Carl Zeiss, Germany). Images were acquired using a 20x objective (Plan-Apochromat 20x/0.8 M27) with a pixel size of 120 nm. The following excitation lasers were used: 405 nm (DAPI), 488 nm (Atto-488), 561 nm (Cy3) and 639 nm (AF-647).

Uncleared ITTs were imaged using ZEISS Axio Observer Z1 and TriMScope II Multiphoton microscope (Miltenyi Biotec). Two-photon images (Fig 2) were acquired using a pulsed Titanium:Sapphire laser at 800 nm excitation wavelength. Emission signal was detected using PMTs (second harmonic generation (SHG) at 400 nm, BrdU-GFP at 525 nm, autofluorescence at 600 nm).

## Light sheet fluorescent microscopy (LSFM) imaging

Samples were mounted in a cuvette filled with BABB solution and oriented perpendicular to the light sheet path for LSFM imaging. Emission was collected using 4× (NA 0.35, MI PLAN, DC57 WD16 O1 dipping cap) and 12× (NA 0.53, MI PLAN, DC57 WD10 O1 dipping cap) objectives, both corrected for the BABB refractive index (n = 1.56). Optically cleared ITTs (Fig 3) were imaged using UltraMicrosocpe II Super Plan (Miltenyi Biotec) installed with ImspectorPro software (version 7.1.15). Samples were imaged in BABB solution using 4x and 12x objective lenses (4x NA 0.35 MI PLAN objective with

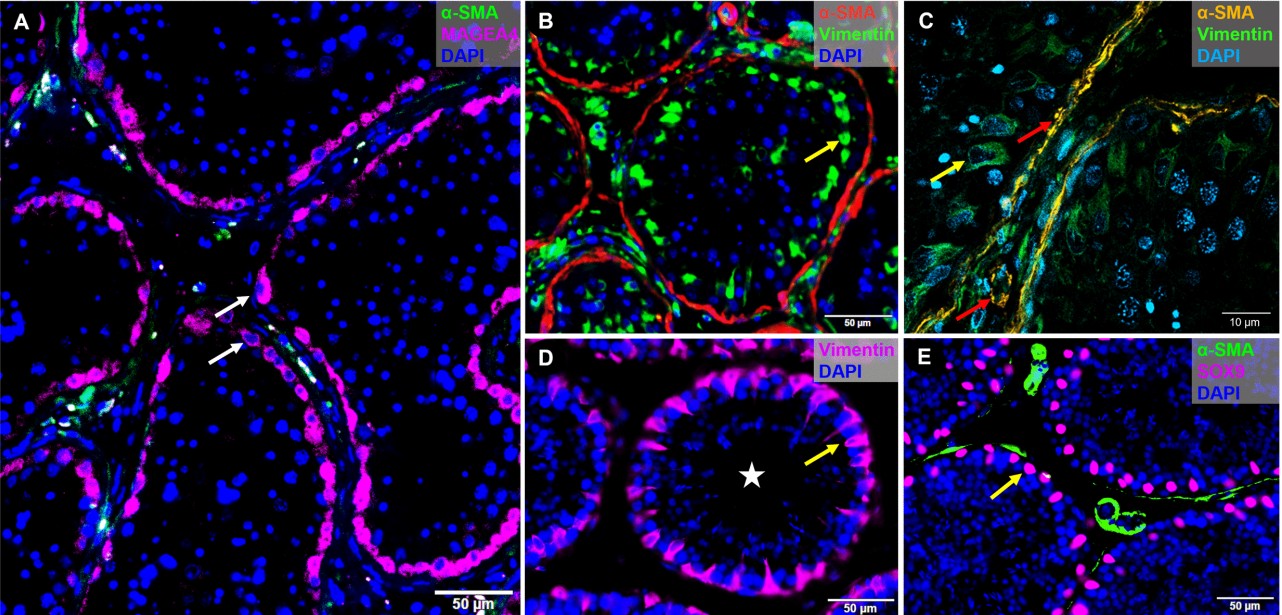

**Fig 1. Immunofluorescent staining of testicular germ and somatic cell markers.** Human testicular tissue images acquired with fluorescent microscope showing **(A)** MAGEA4-positive spermatogonia (white arrows), **(B)** Vimentin-positive Sertoli cells (yellow arrow) and **(C)** zoom-in confocal microscopy showing vimentin-positive cells (yellow arrow) and α-SMA-positive PTMCs (red arrows). Macaque testicular tissue staining of Sertoli cells (yellow arrows) with Vimentin **(D)**, and SOX9 **(E)**. Both were acquired with wide-field fluorescent microscope. Scale bars: 10 and 50 μm. Spermatogonia, white arrows; Sertoli cells, yellow arrows; PTMCs, red arrow; tubular lumen; white star.

DC57 WD16 O1 dipping cap; 12x NA 0.53 MI PLAN with DC57 WD10 O1 dipping cap). Fluorescence was excited using 3 light sheets from either one or two sides (40–70% of available width, light sheet thickness: 3.86 μm) using a white light laser (SuperK Extreme, NKTPhotonics). Emission signal was detected on a sCMOS camera (pco.edge 4.2, pixel size, 6.5 μm) using dynamic focus and an exposure time of 200 ms resulting in an effective pixel size of 1.625 μm for 4x and 0.542 μm for 12x imaging. Emission and excitation filter settings for fluorophores were as follows: Atto488 (Excitation: BP470/40 nm, Emission: BP525/50 nm), Cy3 (Exc: BP 546/10, Em: BP577/25), AlexaFlour 647 (Exc 640/20, Em 690/50) and NucSpot 750/780 (Exc 710/75, Em 810/90 nm). Z-stacks were acquired with a step-size of 2.5 or 5 μm. Mosaic image acquisition was done on large samples by generating multiple tiled stacks that were later stitched to align and fuse adjacent stacks thereby reconstructing full images. We generated multiple tiled stacks with a 20% overlap. Tiled images were pre-aligned in Zeiss Arivis Pro TileSorter using grid mode by specifying the number of columns and rows, the acquisition order ("Straight, Rows"), and the image overlap (20%). To ensure accurate registration, the alignment was manually adjusted based on prominent features, such as αSMA-positive blood vessels, throughout the volume. Once the alignment was satisfactory, the tiles were stitched to generate the final mosaic images.

## Image analysis

Datasets from microtome sections were analyzed using Qupath or Olyvia, allowing us to capture still images (Fig 1). Uncleared ITT images were analysed using Olyvia_V291_B1377, Qupath-0.4.4, ImageJ-1.54g and Omero (OMERO.web 5.25.0). Light sheet imaging data were processed into Arivis files using the Arivis SIS Converter (Version 3.5.1) for visualization in Zeiss arivis Pro (Version 4.1.1, Zeiss), which supports large image volume analysis. Additionally, datasets from LSFM were also analyzed using ImageJ-1.54g. From these datasets, 2D images and 3D videos were generated

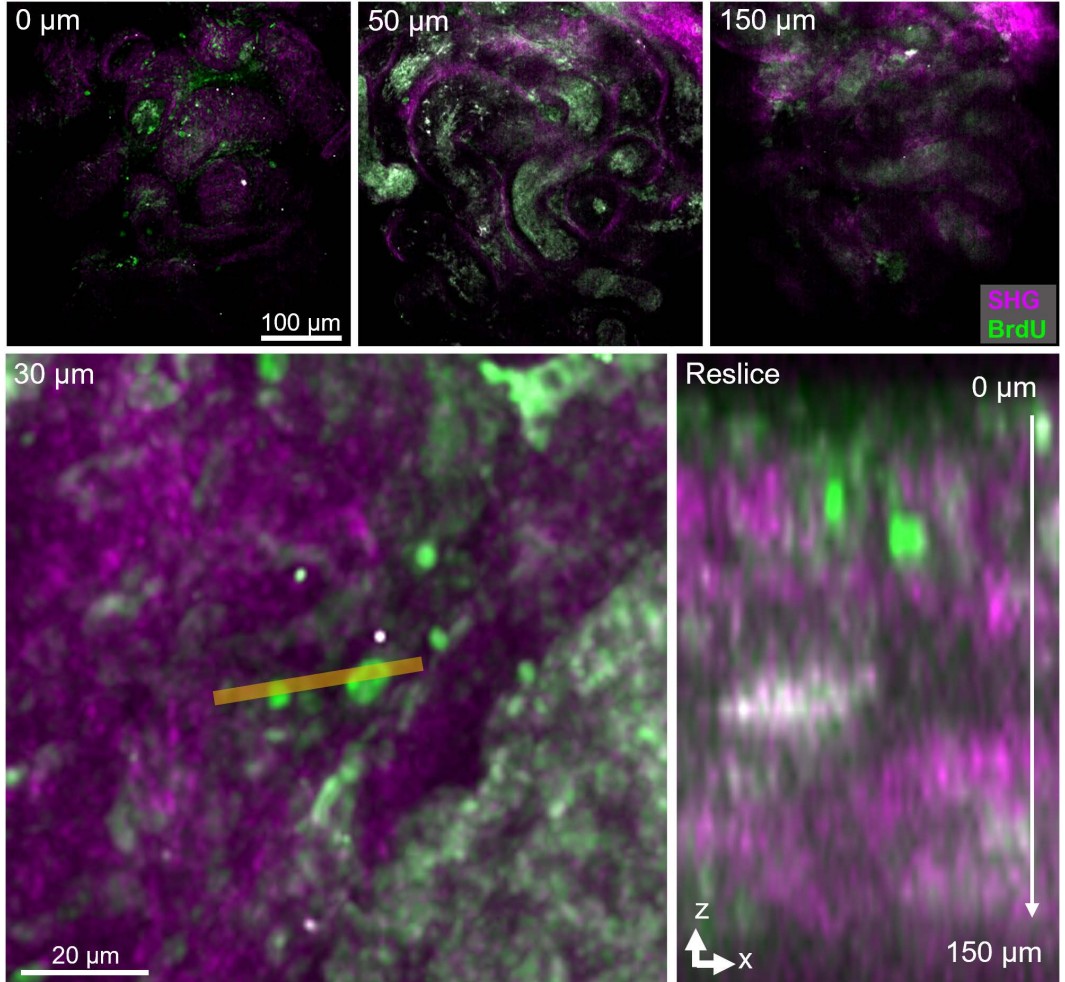

**Fig 2. Two-Photon imaging of uncleared ITTs from marmoset testis.** Decreasing signal-to-noise ratio at different depths of image stack (0 μm represents the top of the tissue). Zoom-in and reslice along the xz-axis at the indicated orange line. The acquired testicular tissue image had pixel size of 0.67 x 0.67 x 3.00 μm (X x Y x Z). Scale bars, as indicated. SHG, second harmonic generation.

(Figs 4-7, S1 Fig and S1-S4 Movies). We used the movie exporter to create MPEG4 movies at a resolution of 1280x720 and 25 frames per second (S1 Movie). Volume renderings of complete tissue blocks (Figs 5B, 6A-6B and 7D-7F) using 4D viewer offered deep tissue visualization thereby enabling creation of storyboards that were exported as movies (S1-S4 Movie). A notable feature of the 4D viewer was its ability to clip thick tissues along the X, Y, and Z planes, aiding visualization of germ and somatic cells in the dense monkey tissues. Achieving optimal settings was a prerequisite to consider whole mount immunolabelling and optical clearing successful.

Fluorescence signal was quantified across tissue depth using Fiji version 1.54 [20]. At each depth (0–1500 μm, 250 μm steps), 20 regions were manually selected at sites of α-SMA–Cy3 staining outlining the testicular tubules. Square ROIs (10 × 10 pixels) were generated around each point, and corresponding background ROIs were obtained by translating them into the tubule interior (unstained region). Mean intensity and standard deviation were measured for each ROI. Signal-to-noise ratio (SNR) was calculated as SNR= (I signal -I background)/$\sigma$_background and signal-to-background ratio (SBR) as SBR = (I signal)/I background. YZ reslices were created along a horizontal line using Fiji.

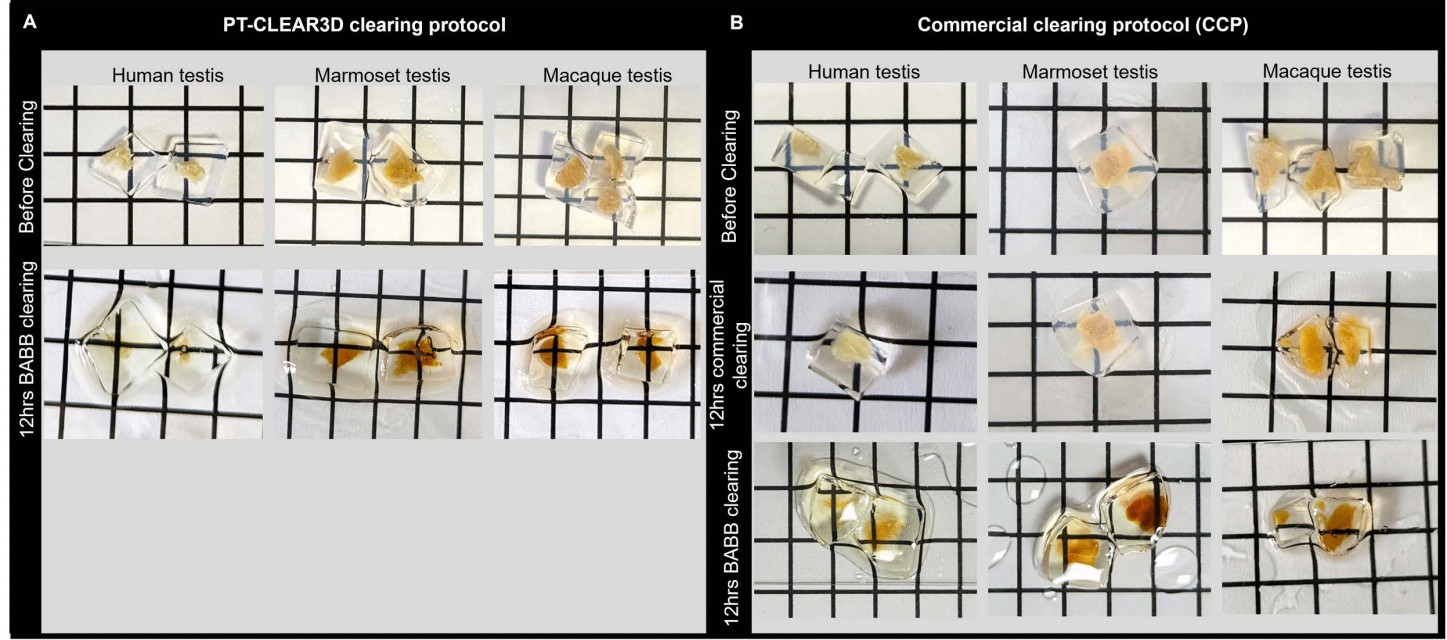

**Fig 3. Images showing primate ITTs before and after clearing comparing PT-CLEAR3D and a commercial kit.** Clearing was considered efficient if the grids below the tissues were visible. **(A)** PT-CLEAR3D clearing in BABB. **(B)** Commercial kit clearing followed by BABB. Several (2-4) testicular tissues were processed for each species in one vial and therefore, images may not represent the same testicular tissue before and after clearing. BABB: Benzyl alcohol/benzyl benzoate.

## Results

### Qualitative analysis of the primate testes using classical methods

Traditional approaches for studying testicular tissue architecture and cell dynamics include histological sectioning and immunofluorescence staining of teased tubules. These methods have been instrumental in revealing key aspects of testis structure, including the organization of seminiferous tubules and the clonal expansion of spermatogonial stem cells (SSCs) [10]. The use of specific cellular markers allow multi-parametric acquisition of human, marmoset and macaque testis at high specificity and resolution (Fig 1). Tubular and interstitial regions were morphologically intact as well as the epithelial organization of the seminiferous tubules in the monkey ITTs. This was indicated by MAGEA4-positive spermatogonia (Fig 1A), SOX9- and Vimentin-positive Sertoli cells and alpha-SMA expression in peritubular myoid cells (Fig 1B-1E) and the vasculature (Fig 1C and 1E). Analysis of the marmoset and macaque testes revealed architecture of normal male adults with highly populated seminiferous epithelium. Additionally, there was predominant basal localization of MAGEA4-positive spermatogonia and SOX9-positive Sertoli cells, indicating maturity of the testis. This was accompanied by a strong expression of SMA with a thin layer of peritubular myoid cells (PTMCs) and the presence of lumen in the seminiferous tubules across all species. On the contrary, the human testes revealed a pathological condition with loose testicular parenchyma and irregular distribution of germ cells in the tubules.

The reconstruction of 2D microtome sections to evaluate clonal expansion of SSCs and the associated testicular cells is rather challenging. Therefore, in our attempt to maintain the tissue integrity, uncleared intact ITTs (0.5 to 10 mm$^3$) were imaged using a confocal microscope and a 2-photon microscope. Although these platforms enabled deeper penetration into the tissue, imaging was only possible up to 150 μm (Fig 2). The imaging depth-limitations imposed by the natural

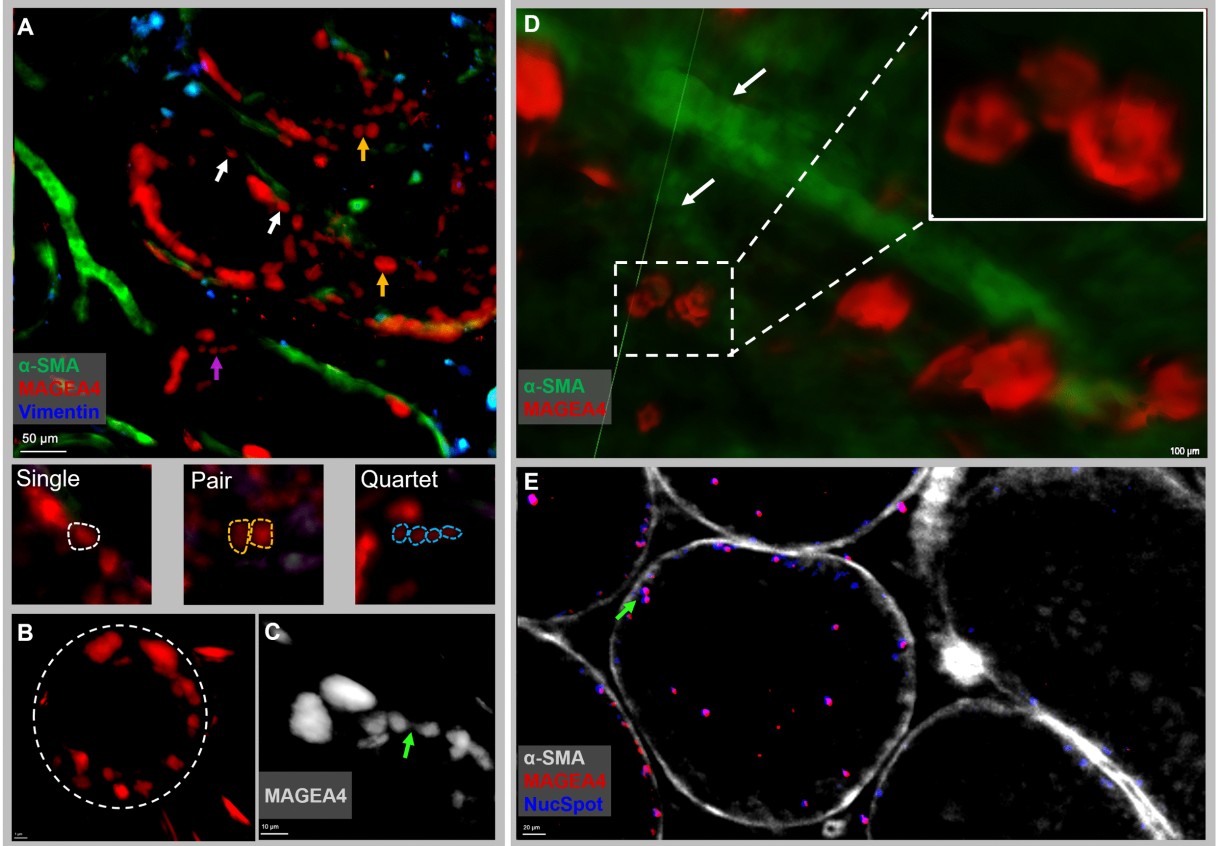

**Fig 4. Light sheet 2D images of intact testicular tissues after processing with PT-CLEAR3D protocol. (A)** Representative tubules with surviving germ cell clones in a human transgender model: single germ cells, white arrows; pairs, yellow arrow; and quartets, magenta arrow (Scale bar: 50 μm). **(B)** A single tubule with germ cell clones (Scale bar: 1 μm) and **(C)** a cellular bridge connecting a pair of MAGEA4-positive spermatogonia (Scale bar: 10 μm). Human testicular dimensions: 1 mm x 1 mm x 1 mm {width (X) x height (Y) x depth (Z)} with 1 mm³ volume. **(D)** Germ cell clone of four spermatogonia (inset) in marmoset testicular tissue in close association with the blood vessel (white arrows) and a cellular bridge **(E)**. Tissue volume was 29 mm³ with 3.2 mm x 3.2 mm x 2.8 mm {width (X) x height (Y) x depth (Z)}. Scale bars: 100 and 20 μm.

scatter of biological non-cleared samples prompted the development of an optical tissue clearing strategy including multi-parametric image acquisition using LSFM.

In summary, the results obtained from ultra-thin 2D microtome sections and uncleared ITTs prove that only a limited amount of information is collected from large tissue volumes, thereby validating our approach to using tissue clearing techniques. This approach for deep tissue imaging of ITT aims to offer a toolbox for imaging large testicular tissue volumes and subsequently further our understanding of testicular development in primates.

## Evaluation of 3D testicular structure in optically cleared ITTs

Most biological tissues have an intrinsic opaque appearance caused by light scattering reflected off molecules, membranes, organelles, and cells in the tissue that impedes acquisition of sharp images [4]. This challenge of light scattering has been resolved by RI matching using clearing agents such as BABB. To establish an efficient optical clearing strategy for use with the primate testis and that retains tissue integrity, we compared two immunolabelling and clearing protocols: a commercial kit (S2 - S5 Figs) and PT-CLEAR3D (in-house) (S6 – S9 Figs).

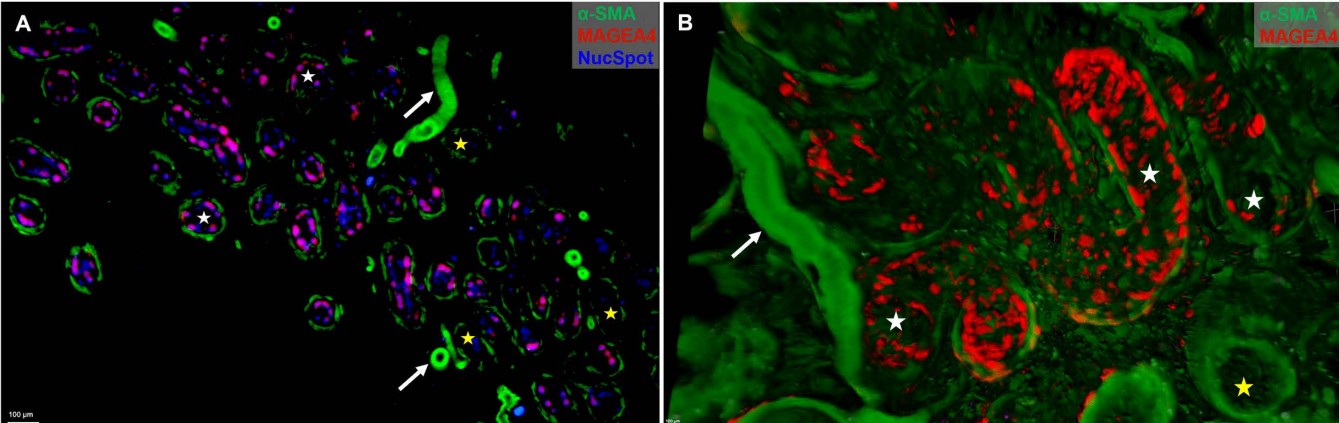

**Fig 5. Testicular structure showing seminiferous tubules with and without spermatogonia.** Tubules with surviving spermatogonial (white stars) clones and without spermatogonial clones (yellow stars) following spermatogenesis suppression in transgender models. Optically cleared with the PT-CLEAR3D protocol and acquired with Ultra Super Plan II LSFM. **(A)** A single optical plane of 3D reconstruction of testicular cells showing seminiferous tubules with surviving spermatogonial clones (MAGEA4-positive), empty tubules, basement membrane and vasculature (SMA), white arrows. Tissue dimensions: 3 mm x 3 mm x 4 mm {width (X) x height (Y) x depth (Z)} totaling 36 mm$^3$. **(B)** Volumetric projection showing persisting spermatogonial clones in one seminiferous tubule (white stars) while the neighboring tubules are empty with no visible MAGEA4-positive spermatogonia (yellow star). Testicular dimensions: 1 mm x 1 mm x 1 mm {width (X) x height (Y) x depth (Z)} with 1 mm$^3$ volume. Scale bars: 100 µm.

Optical clearing with PT-CLEAR3D enabled efficient immunolabelling and visualization of whole mount testicular tissue architectures in human, marmoset, and macaque samples across depths up to 2 mm (Fig 3a, S6 - S9 Figs). Images remained sharp throughout the entire depth analyzed in all three species; no blurring was observed (S6 – S8 Figs). Signal-to-noise ratio (SNR) and signal-to-background ratio (SBR), measures indicative of image quality and contrast respectively, decreased slightly throughout the analyzed depth range in human tissues, exhibiting higher values than observed in marmoset and macaque samples (S9 Fig). In marmoset and macaque tissues, SNR and SBR remained consistent throughout the analyzed depth range – although at lower levels compared to human samples (S9 Fig). This enhanced efficiency in human tissues potentially resulting from GAHT-induced impairment of spermatogenesis causing degradation of testicular tissue density, thereby facilitating greater penetration of antibody labelling and clearing reagents, and less tissue scattering.

For all species, testicular germ and somatic cells, including the vasculature, were observed in an undisturbed environment, thus maintaining spatial organization and cellular network. Most importantly, visualization of the ITTs was possible at both tissue and cellular resolution (Figs 4 - 6). Testing the commercial clearing kit demonstrated robust ITT antibody labelling, however, tissue clearing proved inefficient, evidenced by persistent opacity when assessed on grids (Fig 3B). LSFM images revealed progressively blurry structures with increasing depth, accompanied by decreasing signal-to-noise ratio (SNR) and signal-to-background ratio (SBR) (S2 – S4 Figs). Subsequent incubation in BABB (12 hours) rescued clearing efficiency, achieving complete transparency as confirmed by grid visibility through the tissue, enabling subsequent imaging (Fig 3B). Similarly, visual blurriness over depth was decreased, and SNR and SBR was rescued after incubation with BABB (S2 – S4 Figs). It should be noted that image acquisition settings – including laser power and orientation – were not necessarily identical across all comparisons.

In conclusion, our results demonstrate that PT-CLEAR3D consistently outperformed the commercial kit in achieving complete optical clearing of primate testis tissue. While the commercial kit offered adequate antibody labelling, its inefficiency in clearing larger testis volumes necessitated post-treatment with BABB for effective imaging. It is important to note that all protocols described herein were specifically developed and validated for use in primate testis tissue; their

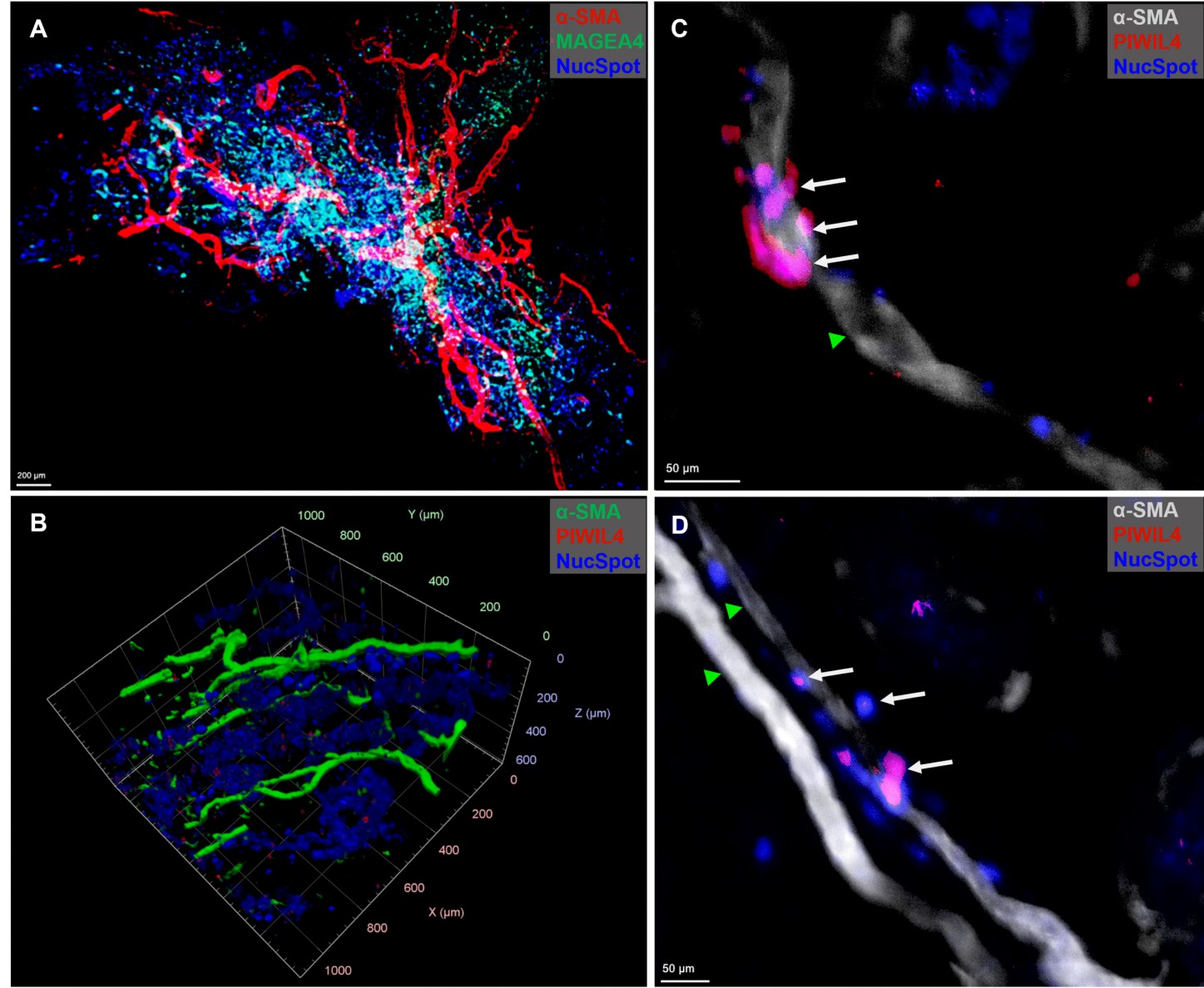

**Fig 6. Spatial organization of human testicular tissue in a transgender model.** LSFM microscopy of optically cleared (PT-CLEAR3D) testicular tissues. **(A)** Optically cleared 50 mm³ sized testicular tissue showing association of testicular germ, somatic cells and vasculature. Scale bar: 200 μm. Tissue dimensions: 3 mm x 3 mm x 4 mm {width (X) x height (Y) x depth (Z)} totaling 36 mm³. **(B)** Spatial organization of the least differentiated spermatogonia (PIWIL4-positive) and vasculature (SMA). Tissue dimensions: 1 mm x 1 mm x 0.6 mm {width (X) x height (Y) x depth (Z)} totaling 0.6 mm³. **(C)** and **(D)** Close association of PIWIL4-positive spermatogonia, white arrows, with the blood vessel (green arrowheads). Scale bars: 50 μm.

applicability to other organ systems remains unexplored. Therefore, the detailed findings presented hereafter were generated using our optimized, in-house protocol – PT-CLEAR3D – providing a reliable method for high-resolution 3D visualization of testicular structures.

Our results showed a few seminiferous tubules populated with surviving MAGEA4-positive spermatogonial clones and occasional Vimentin-positive Sertoli cells in the human transgender model (Fig 4A-4C) post-GAHT. Notably, some of the neighbouring tubules had neither MAGEA4-positive spermatogonia nor Vimentin-positive Sertoli cells (Figs 4A and 5B),

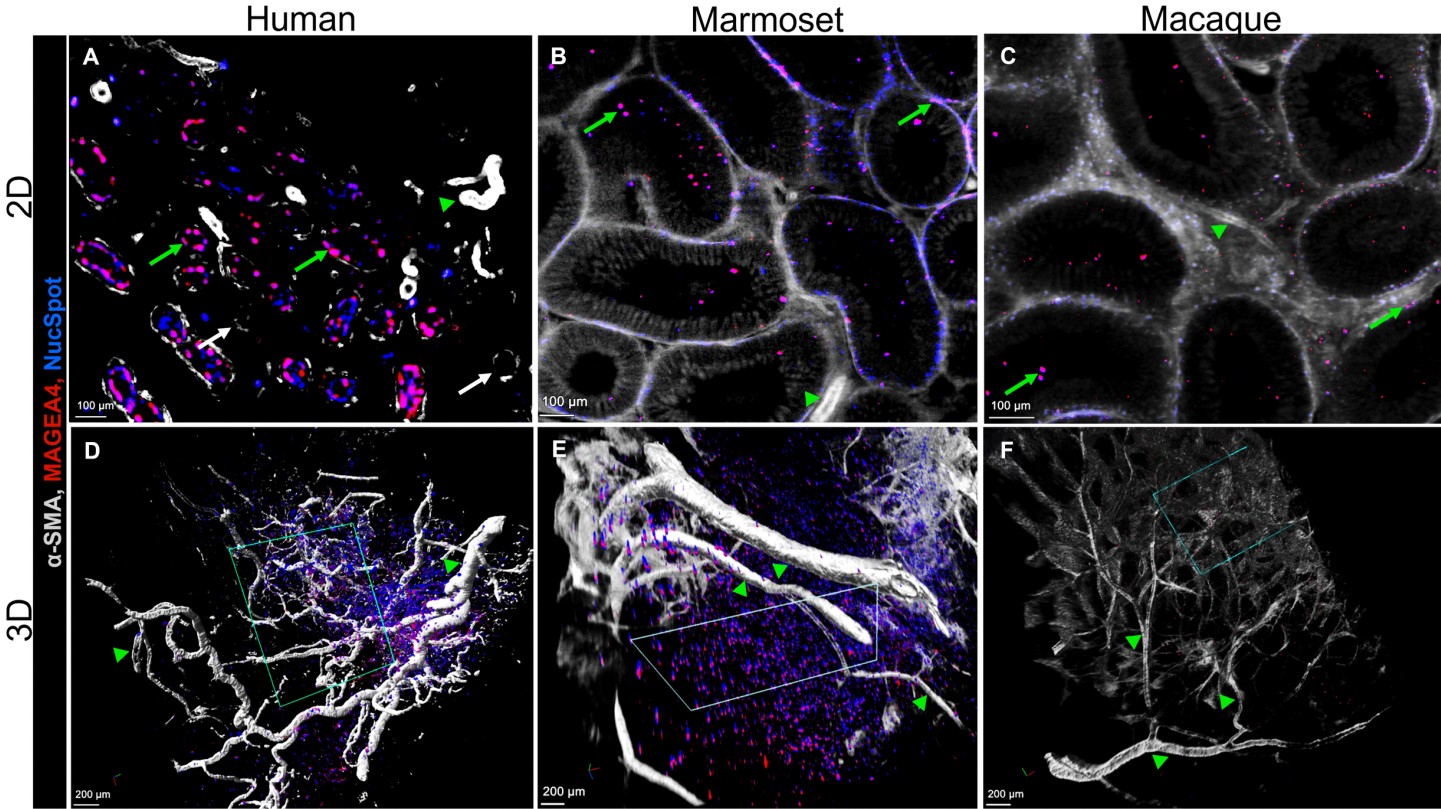

**Fig 7. Large-volume imaging of intact and optically cleared testicular tissues across three primate species. Top row (A-C):** 2D visualization of the testes, **(A)** Adult transgender model showing basal localization of MAGEA4-positive spermatogonia (green arrows) in the tubules while some tubules have no MAGEA4-positive spermatogonia (white arrows); Tissue dimensions: 3 mm x 3 mm x 4 mm {width (X) x height (Y) x depth (Z)} totaling 36 mm³; **(B)** marmoset, tissue dimensions: 3.2 mm x 3.2 mm x 2.8 mm {width (X) x height (Y) x depth (Z)} totaling 29 mm³ and **(C)** macaque testis showing MAGEA4-positive spermatogonia in the basal and adluminal compartments. Tissue dimensions: 3.2 mm x 3.2 mm x 2.4 mm {width (X) x height (Y) x depth (Z)} totaling 25 mm³. Green arrowheads: vasculature. Scale bars: 100 µm. **Bottom row (D-F):** 3D visualization of the primate testes focusing on the vasculature (green arrowheads) in volumetric projections of **(D)** adult human transgender model showing spiralling phenomenon and vascular branching, and in normal breeding adult **(E)** marmoset and **(F)** macaque showing straight configuration of vasculature. The green rectangles are positions of the 2D images in 3D projections. Colour settings are kept constant in 2D and 3D. Scale bars: 200 µm.

revealing the impact of GAHT leading to testicular pathology. An in-depth analysis of the surviving spermatogonial clones and associated somatic cells was possible at both the tissue and cellular levels. Most of the MAGEA4-positive spermatogonia were localized at the basal compartment of the seminiferous epithelium. Interestingly, singlets, pairs or quartets were observed in the ITTs (Fig 4A); moreover, as a result of the 3D reconstruction of the cells in ITTs, intricate details such as cellular bridges were revealed as depicted in Fig 4C. Similar observations were made in the monkey ITTs where clones of spermatogonia were clearly visible (Fig 4D), including cellular bridges (Fig 4E), offering a valuable development for downstream analysis. In addition to the MAGEA4-positive spermatogonia occurring as single cells, pairs or quartets in the human testis, groups of eight cells were observed in the monkey ITTs (S1D Fig). The visualization of the spatial organization of the primate testis was aided by 2D and 3D images and movies of ITTs from the LSFM datasets that were acquired with 4x (Fig 5A, S1 Movie) and 12x (Fig 5B, S2 Movie) objective lenses. In Fig 5A, testicular cell distribution is appreciated clearly using the stack of images acquired from cleared human ITT. Fig 5B shows one seminiferous tubule populated with MAGEA4-positive spermatogonia that persisted despite being subjected to GAHT over an extended period to suppress spermatogenesis. The tubule is convoluting in and out of the tissue surface and could be tracked along its

length for further analysis using Zeiss Arivis Pro software (S2_Movie). The analysis revealed that what would otherwise be quantified as several tubes in 2D microtome sections is indeed one tubule, as seen in 3D visualization. This significant development will help overcome the long-standing limitations of classical histology and allow analysis of more enormous tissue volumes, as seen in Fig 6A and 6B and supplementary (S1-S4 Movies). The image on Fig 6A represents an optically cleared human ITT, 50 mm³ tissue. It shows a broad distribution of the blood vessels and the associated cells - MAGEA4-positive spermatogonia and other testicular cells.

Comparable to MAGEA4-positive spermatogonia, PIWIL4-positive spermatogonia, which are the least differentiated, were predominantly localized at the basement membrane of the ITTS, but they occurred rarely compared to MAGEA4-positive spermatogonia across the three species. Notably, some PIWIL4-positive clones were closely associated with the vasculature in the human testis, occurring along, around, and between blood vessels (Fig 6C and 6D). The human transgender testes offers a characteristic model for aregressed testis. Similarly, the distribution of MAGEA4-positive spermatogonia was predominantly localized at the basal compartment in the marmoset and macaque ITTs. However, clones of MAGEA4-positive spermatogonia were also observed in the adluminal compartment. Unlike human ITTs, no tubules with depleted germ cells were observed in the monkey ITTs. Nuclear staining indicated densely populated tubular and interstitial regions, as evidenced by Vimentin-positive Sertoli cells, further confirming the status of unperturbed testes. A strong expression of SMA accompanied this observation with a thin layer of PTMCs and the presence of lumen in the seminiferous tubules.

Upon comparing 2D and 3D images of the optically cleared ITTs (Fig 7), there is no doubt that PT-CLEAR3D revealed intricate details of the testicular architecture that microtome serial sections would struggle to achieve. The 3D reconstruction shows the testicular spatial organization more clearly than microtome sections (Fig 7A-7C). Given the dense nature of a normal primate testis, the blood vessels provided an excellent anatomical landmark to analyse the datasets. A single layer of thin endothelial cells forms the vasculature that stained positive for alpha-SMA similar to the PTMCs that surround the tubules. For this reason, obtaining optimal fluorescent signal for both the vasculature and basement membrane concurrently was challenging since they were acquired with the same channel and resolution. Consequently, the vasculature had increased signal intensity compared to the basement membrane. It is worth noting that the 2D images are single optical sections and thus, there was no projection of data whereas the 3D images are image projections generated from the data sets. Maximum projections were rendered in 3D and subsequently, volumetric and maximum intensity were used for image analysis.

Taken together, the 3D visualization of testicular architecture is a valuable tool that uncovers rare occurrences within the testis and offers significant potential for further analysis of spermatogenesis kinetics. This breakthrough opens new avenues of research in testis biology. Our study adds to the existing literature on tissue clearing techniques with a special focus on reproductive biology, and highlights the method's versatility for use in bicompartmental organs.

## Testicular vasculature network assessment in optically cleared ITTs

The vasculature is associated with niche formation based on the localization of the undifferentiated spermatogonia and the accompanying cells, which subsequently influence niche plasticity in the mouse [21]. Alterations in the vasculature pattern led to the rearrangement of the spermatogonia. The whole mount staining and optical tissue clearing elegantly revealed complex and intricate vasculature networks in the ITTs from all primate species (Fig 7). The testicular vasculature is instrumental in regulating the temperature and is closely related to the seminiferous tubules and the Leydig cells in the interstitial space [8].

Our findings reveal abundant and highly branched vascular networks in the human transgender model compared to the monkey models. The intratesticular blood vessels were observed running between the tubules, which allows for determining their diameter based on the 2D images. With microtome sections, blood vessels appear as circular objects and analysis of parameters such as length and branching is impossible. Although the 2D images of optically cleared ITTs

                                    

allowed reconstruction of the testicular cells and subsequently showed elements such as the length and width of the blood vessels, these parameters are more precise when analysed using 3D images (Fig 7D-7F and S1A-S1C Fig).

Notably, the vasculature (testicular artery) in the human transgender models showed spiralling configurations, while the normal breeding monkey ITTs exhibited straight configurations. The spiralling phenomena of the intratesticular arteries mainly reduce blood flow to the delicate seminiferous tubules [22]. Vascular networks comprising large and small vessels were detected in all tissues, including their distribution across the ITTs. Large vessels branched as they went deeper into the tissue to maintain testicular function. Overall, the intricate vasculature network maintains the much-needed high metabolic activity in the seminiferous tubules through the efficient supply of nutrients and oxygen, and any disturbances in the blood supply could cause testicular malfunction. It is evident that testicular vasculature plays a key role in the delicate balance between cell proliferation, differentiation, and apoptosis. Our findings provide a foundation for advanced analysis, such as branching points of the vasculature, influence of the niche and possibly stemness. Moreover, species-specific differences in the vasculature networks may exist, prompting future extensive studies to evaluate the testicular vasculature using our newly developed toolbox.

## Discussion

We developed PT-CLEAR3D, a cutting-edge toolbox for 3D visualization of intact primate testes, enabling testicular structure and function analysis while preserving tissue integrity. To our knowledge, this is the first innovative approach to examine successfully the spatial organization and cellular associations in intact primate testes. First, we compared our in-house protocol, PT-CLEAR3D, with a commercial kit. PT-CLEAR3D outperformed the commercial kit for immunolabelling and tissue clearing, achieving effective antibody labelling and tissue clearing. We performed signal-to-noise and signal-to-background measurements that indicated that samples subjected to double processing with CCP and BABB had decreasing fluorescence intensity. This reduction is likely due to fluorophore degradation and quenching effects caused by prolonged chemical exposure, as well as the time elapsed between the first and second imaging sessions (approximately one week). With regard to labeling specificity, staining was generally more specific in human tissue, which showed higher signal intensity and lower background levels. In non-human primate tissue, signal intensity was overall lower and background higher; however, labeling appeared consistent across depth.

Next, we evaluated various staining and imaging platforms for studying the primate testis. While 2D microtome sections allow for clear identification of germ and somatic cells, they are inherently limited in depth, precluding 3D reconstructions. Further, whole mount staining of uncleared ITTs allowed 3D visualization but was limited in imaging depth and the appreciation of germ cell clones using confocal and two-photon microscopy. Optical clearing through PT-CLEAR3D emerged as an elegant solution, facilitating the visualization and analysis of intact testes without compromising their structural integrity. Additionally, PT-CLEAR3D enabled 3D cellular reconstruction and visualization of cellular associations, including cellular bridges and associated vasculature, resolving the challenges posed by traditional microtome sections and uncleared ITTs. Lastly, we assessed immunolabelling and clearing in Bouin's and PFA fixed ITTs in humans, marmosets, and macaques. PT-CLEAR3D was effective across all species and both fixatives. Its versatility show great promise for multispecies applicability and compatibility with various tissue fixatives and fluorophore multiplexing in the field of reproductive biology.

Traditional approaches, such as histological sectioning and immunofluorescence staining of 2D sections, have provided valuable insights into testicular architecture and the clonal dynamics of spermatogenesis. However, Clermont [9] pointed out several limitations of 2D sections that also apply, albeit to a lesser extent, to whole mount preparations. A significant drawback of classical histology is the distortion that occurs in individual sections due to tearing, folding, and compression, which hampers volumetric reconstruction [4]. Additionally, assessing mixed cellular interactions in microtome sections is challenging; even though close spatial localization is used to determine cells that are in synchrony and in the same division stage during processes like mitosis [10]. Our use of PT-CLEAR3D and 3D reconstruction revealed rare features, such as cellular bridges, highlighting the advantages of deep volumetric imaging. Furthermore, microtome sections may miss certain

generations of germ cells, a limitation that whole mount preparations partially address, though achieving sufficient depth of visualization remains a challenge [23,24]. Our work on regressed testis samples from a human transgender model also confirmed that the same tubule is often evaluated multiple times. Lastly, the pressure exerted on delicate tubules during biopsy collection can displace germ cells and disrupt the epithelium, leading to distorted cellular associations [9]. While our study could not definitively confirm this assertion, PT-CLEAR3D shows significant promise for future work with whole primate testis samples. Certainly, optical clearing techniques are gaining attraction across various fields despite some downsides. For example, organic solvent-based methods like BABB can effectively clear tissue, but they often cause sample shrinkage [5]. Therefore, necessary adjustments should be considered when determining the actual size of intratubular and extratubular components. Noteworthy, PT-CLEAR3D that includes BABB as the clearing step was tested against only one commercial kit using the primate testes. Its compatibility with other clearing protocols and in other organs was not tested in this work.

The imaging depth of multiphoton microscopy limited its use in assessing uncleared ITTs in the current study. However, it effectively identified spermatogenic stages in fresh, intact tissue without the need for clearing or cellular markers in a rodent model [25]. In this rodent model, minimal DNA fragmentation was observed following exposure to different laser intensities. These findings indicate the potential of multiphoton microscopy for real-time, label-free visualization of spermatogenesis in clinical settings, such as testicular sperm extraction. For fixed tissues, optical clearing mitigates the challenges posed by opaque samples [4]. Still, it requires complete penetration of the staining reagent [5], necessitating the optimization of incubation periods, particularly for the dense primate testes. In our analysis of human ITTs following GAHT, we observed loose testicular parenchyma, seminiferous tubules depleted of germ and Sertoli cells and blood vessels exhibiting spiralling configurations compared to monkey ITTs. As noted in this study, the loss of germ cells may result in testicular compaction and increased blood vessel density due to the reduced testis size [26].

Humans and marmosets have multi-tubular arrangements with multiple spermatogenic stages per cross-section, unlike macaques, where these stages are separated longitudinally [19,27]. The non-destructive analysis of large samples combined with high resolution of LSFM could facilitate the study of such species-specific differences. Our findings show similar staining patterns for MAGEA4-positive and PIWIL4-positive spermatogonia in the ITTs of humans, marmosets, and macaques. We observed fewer PIWIL4-positive spermatogonia, the least differentiated, compared to MAGEA4-positive spermatogonia across all species. Notably, our novel toolbox allowed us to visualize clones of PIWIL4-positive spermatogonia closely associated with blood vessels in the human testis. These findings highlight the value of the transgender model, alongside monkey tissues and PT-CLEAR3D technology, in the study of testicular physiology and pathophysiology.

## Conclusion

Overall, the PT-CLEAR3D toolbox is an efficient and cost-effective resource for investigating testicular architecture and function, revealing rare occurrences in the testis in both health and disease. Its versatility allows the use of various fixatives and fluorophores, enabling researchers in the reproductive field—and in the study of other bicompartmental organs—to examine and compare spermatogenesis kinetics across multiple species. By employing specific cellular markers and utilizing diverse immunolabelling techniques and imaging platforms in three primate species, this study effectively addresses the challenges of traditional histology and the whole mount preparation of teased tubules, thus opening new avenues for research in testis biology.

## Supporting information

**S1 Movie. 3D view of human testicular tissue.** The movie shows seminiferous tubules with surviving spermatogonial clones, empty tubules, basement membrane and vasculature. Supplementary to Fig 4A. Tissue dimensions: 3 mm x 3 mm x 4 mm {width (X) x height (Y) x depth (Z)} totaling 36 mm$^3$. Scale bar: 500 μm. S1 – S4 Movie: https://doi.org/10.6019/S-BIAD1898.
(MP4)

**S2 Movie. Video showing different viewpoints of a single tubule with spermatogonial clones.** The tubule is convoluting in and out of the tissue while the neighboring tubules are depleted of spermatogonia. Supplementary to Fig 4B. Testicular dimensions: 1 mm x 1 mm x 1 mm {width (X) x height (Y) x depth (Z)} with 1 mm³ volume. Scale bar: variable with the play option.
(AVI)

**S3 Movie. Spatial organization of a 50 mm³ sized human testicular tissue in a transgender model.** Supplementary to Fig 5A. Tissue dimensions: 3 mm x 3 mm x 4 mm {width (X) x height (Y) x depth (Z)} totaling 36 mm³. Scale bar: 100 µm.
(MP4)

**S4 Movie. Spatial distribution of least differentiated spermatogonial (PIWIL4-positive) and vasculature (SMA).** At time point 00.03.84 to 00.04.76 the focus is on a clone of PIWIL4-positive spermatogonia in close contact with the blood vessel. Supplementary to Fig 5B. Tissue dimensions: 1 mm x 1 mm x 0.6 mm {width (X) x height (Y) x depth (Z)} totaling 0.6 mm³. Scale bar: variable with the play function.
(AVI)

**S1 Fig. 3D images of the vascular network and spermatogonial clones.** Vascular network (green) in close association with MAGEA4-positive spermatogonia (red) and other testicular cells (blue) in volumetric projections in **(A)** human transgender model, tissue dimensions: 3 mm x 3 mm x 4 mm {width (X) x height (Y) x depth (Z)} totaling 36 mm³, **(B)** marmoset, tissue dimensions: 3.2 mm x 3.2 mm x 2.8 mm {width (X) x height (Y) x depth (Z)} totaling 29 mm³, and **(C)** macaque, tissue dimensions: 3.2 mm x 3.2 mm x 2.4 mm {width (X) x height (Y) x depth (Z)} totaling 25 mm³. The cells that appear magenta in color are spermatogonia that are stained positive for both MAGEA4 and nucleardye. Scale bars: 50 µm. **(D)** Macaque ITT showing distribution of testicular cells. Clones of MAGEA4-positive spermatogonia were observed occurring as single cells, orange circle; in pairs, cyan circle; or in groups of four, green circle or eight cells (not shown) as marked by the circles. **(D1)** Inset of a group of six spermatogonia, notice that spermatogonia marked with green arrows have been sliced during collection of the tissue. Scale bars: 200 and 100 µm (inset).
(TIF)

**S2 Fig. Depth-dependent signal quality for different clearing approaches in human testis.** Comparison of commercial clearing kit alone (CCP) with commercial clearing kit followed by BABB clearing (CCP+BABB) in human testes. **(A)** Representative optical sections at 50, 250, 500, 1000, and 2000 µm depth showing α-SMA–Cy3 (green), vimentin–488 (red), and MAGEA4–AF647 (far-blue) channels for both clearing conditions. Grayscale insets display the α-SMA channel alone. YZ reslice along the yellow line in the overview image illustrates α-SMA–positive structures across tissue depth. Samples were illuminated with a single light sheet directed from the right. Scale bar, 200 µm. **(B-C)** Signal-to-noise (SNR) **(B)** and contrast (signal-to-background (SBR)) **(C)** measurements along the z-axis. Mean fluorescence intensity and standard deviation was quantified by selecting SMA–Cy3–positive regions outlining testicular tubules at each depth (0–1500 µm, 250 µm steps). Square ROIs (10×10 pixels) were measured for signal, and corresponding background ROIs were obtained by translation into the tubule interior (unstained region). Shown are SNR and SBR for 20 α-SMA–Cy3–positive regions and corresponding background regions for each z-slice+/- SD.
(TIF)

**S3 Fig. Depth-dependent signal quality for different clearing approaches in marmoset testis.** Comparison of commercial kit clearing alone (CCP) with commercial kit clearing followed by BABB clearing (CCP+BABB) in marmoset testes. **(A)** Representative optical sections at 50, 250, 500, 1000, and 2000 µm depth showing α-SMA–Cy3 (green), vimentin–488 (red), and MAGEA4–AF647 (far-blue) channels for both clearing conditions. Grayscale insets display the α-SMA channel alone. YZ reslice along the yellow line in the overview image illustrates α-SMA–positive structures across tissue depth. Samples were illuminated with a single light sheet directed from the right. Scale bar, 200 µm. **(B-C)**

Signal-to-noise (SNR) **(B)** and contrast (signal-to-background (SBR)) **(C)** measurements along the z-axis. Mean fluorescence intensity and standard deviation was quantified by selecting SMA–Cy3–positive regions outlining testicular tubules at each depth (250–1000 µm, 250 µm steps). Square ROIs (10 x 10 pixels) were measured for signal, and corresponding background ROIs were obtained by translation into the tubule interior (unstained region). Shown are SNR and SBR for 20 α-SMA–Cy3–positive regions and corresponding background regions for each z-slice +/- SD.
(TIF)

**S4 Fig. Depth-dependent signal quality for different clearing approaches in macaque testis.** Comparison of commercial kit clearing alone (CCP) versus commercial kit clearing followed by BABB clearing (CCP+BABB) in macaque testes. (A) Representative optical sections at 50, 250, 500, 1000, and 2000 µm depth showing α-SMA–Cy3 (green), vimentin–488 (red), and MAGEA4–AF647 (far-blue) channels for both clearing conditions. Grayscale insets display the α-SMA channel alone. YZ reslice along the yellow line in the overview image illustrates α-SMA–positive structures across tissue depth. Samples were illuminated with a single light sheet directed from the right. Scale bar, 200 µm. (B-C) Signal-to-noise (SNR) (B) and contrast (signal-to-background (SBR)) (C) measurements along the z-axis. Mean fluorescence intensity and standard deviation was quantified by selecting SMA–Cy3–positive regions outlining testicular tubules at each depth (250–1000 µm, 250 µm steps). Square ROIs (10 × 10 pixels) were measured for signal, and corresponding background ROIs were obtained by translation into the tubule interior (unstained region). Shown are SNR and SBR for 20 α-SMA–Cy3–positive regions and corresponding background regions for each z-slice +/- SD.
(TIF)

**S5 Fig. Depth-dependent signal quality for different clearing approaches across the species. (A)** Signal-to-noise (SNR) and **(B)** and contrast (signal-to-background (SBR).
(TIF)

**S6 Fig. Signal attenuation in PT-CLEAR3D – cleared human testicular tissue.** Overview image at 1000 µm depth (upper left) with zoomed regions (white rectangle) shown at different depths across the stack (right panels) for human testes. All three fluorescence channels are displayed in color (α-SMA, red; MAGEA4, green; NucSpot, blue) and individually in grayscale, with brightness and contrast kept consistent across depths. A YZ reslice along the yellow line in the overview (lower left) illustrates signal distribution through the tissue. Scale bar, 200 µm.
(TIF)

**S7 Fig. Signal attenuation in PT-CLEAR3D – cleared marmoset testicular tissue.** Overview image at 1000 µm depth (upper left) with zoomed regions (white rectangle) shown at different depths across the stack (right panels) for marmoset testes. All three fluorescence channels are displayed in color (α-SMA, red; MAGEA4, green; NucSpot, blue) and individually in grayscale, with brightness and contrast kept consistent across depths. A YZ reslice along the yellow line in the overview (lower left) illustrates signal distribution through the tissue. Scale bar, 200 µm.
(TIF)

**S8 Fig. Signal attenuation in PT-CLEAR3D – cleared macaque testicular tissue.** Overview image at 1000 µm depth (upper left) with zoomed regions (white rectangle) shown at different depths across the stack (right panels) for macaque testes. All three fluorescence channels are displayed in color (α-SMA, red; MAGEA4, green; NucSpot, blue) and individually in grayscale, with brightness and contrast kept consistent across depths. A YZ reslice along the yellow line in the overview (lower left) illustrates signal distribution through the tissue. Scale bar, 200 µm.
(TIF)

**S9 Fig. Signal attenuation in PT-CLEAR3D – cleared human, marmoset, and macaque testis tissue.** Signal-to-noise (SNR) (A) and contrast (signal-to-background (SBR)) (B) measurements along the z-axis. Mean fluorescence intensity

and standard deviation was quantified by selecting SMA–Cy3–positive regions outlining testicular tubules at each depth (250–1500 µm, 250 µm steps). Square ROIs (10 × 10 pixels) were measured for signal, and corresponding background ROIs were obtained by translation into the tubule interior (unstained region). Shown are SNR and SBR for 20 α-SMA–Cy3–positive regions and corresponding background regions for each z-slice +/- SD.
(TIF)

**S1 Table. List of antibodies and reagents used.**
(DOCX)

## Acknowledgments

The authors would like to express their gratitude to Ms. Reinhild Sandhowe-Klaverkamp from CeRA for technical support. Additionally, we appreciate the support of Mr. Jens Wendt (NFDI4BioImage, Münster Imaging Network, University of Münster), who hosted the large datasets and videos for this publication through the BioImage Archive.

## Author contributions

**Conceptualization:** Pauline Wanjiku Kibui, Sarah Weischer, Nils Kirschnick, Thomas Zobel, Stefan Schlatt.

**Data curation:** Pauline Wanjiku Kibui, Sarah Weischer.

**Formal analysis:** Pauline Wanjiku Kibui, Sarah Weischer.

**Funding acquisition:** Stefan Schlatt.

**Investigation:** Pauline Wanjiku Kibui, Sarah Weischer, Nicola von Ostau, Jochen Hess, Nils Kirschnick, Thomas Zobel, Stefan Schlatt.

**Methodology:** Pauline Wanjiku Kibui, Sarah Weischer, Nils Kirschnick.

**Project administration:** Thomas Zobel.

**Resources:** Pauline Wanjiku Kibui, Sarah Weischer, Nicola von Ostau, Jochen Hess, Nils Kirschnick, Thomas Zobel, Stefan Schlatt.

**Software:** Pauline Wanjiku Kibui, Sarah Weischer, Nils Kirschnick.

**Supervision:** Thomas Zobel, Stefan Schlatt.

**Validation:** Pauline Wanjiku Kibui.

**Visualization:** Pauline Wanjiku Kibui, Sarah Weischer, Nils Kirschnick.

**Writing – original draft:** Pauline Wanjiku Kibui, Sarah Weischer.

**Writing – review & editing:** Pauline Wanjiku Kibui, Sarah Weischer, Thomas Zobel, Stefan Schlatt.

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
