## [Decision Letter · Decision Letter 0]

22 Aug 2025

Dear Dr. Schlatt,

Thank you for submitting your manuscript to PLOS ONE. After careful consideration, we feel that it has merit but does not fully meet PLOS ONE’s publication criteria as it currently stands. Therefore, we invite you to submit a revised version of the manuscript that addresses the points raised during the review process.

We look forward to receiving your revised manuscript.

Kind regards,

Su-Ren Chen

Academic Editor

PLOS ONE

“Funding for instrumentation: Zeiss LSM 980: INST 211/898-1 FUGB, Confocal Laser Scanning Microscope.

Miltenyi TriMScope II: INST 211/900-1 FUGB, Multiphoton Microscope

Miltenyi UltraMicroscope II Super Plan: INST 211/899-1 FUGB, Light Sheet Microscope

Deutsche Forschungsgemeinschaft (DFG): CRU326.”

Additional Editor Comments:

Please carefully revise the manuscript according to two reviewers' comments and I am looking forwards to receiving the revision.

Reviewers' comments:

Reviewer's Responses to Questions

**Comments to the Author**

1. Is the manuscript technically sound, and do the data support the conclusions?

Reviewer #1: Yes

Reviewer #2: Yes

2. Has the statistical analysis been performed appropriately and rigorously?

Reviewer #1: N/A

Reviewer #2: N/A

3. Have the authors made all data underlying the findings in their manuscript fully available?

Reviewer #1: Yes

Reviewer #2: Yes

4. Is the manuscript presented in an intelligible fashion and written in standard English?

Reviewer #1: Yes

Reviewer #2: Yes

Reviewer #1: The study introduces an innovative method (PT-CLEAR3D) for tissue clearing and 3D imaging, enabling detailed analysis of spatial organization and cellular interactions within intact primate testes. This approach marks a significant advancement in the investigation of testicular physiology and pathophysiology, overcoming key limitations of conventional histology.

The manuscript is generally well-structured and clearly written, and the methodology is described in detail. Nevertheless, the manuscript would benefit from additional clarification. Below are my comments and revision requests:

1.The authors state (line 406): “However, after subsequently incubating the same ITTs in BABB, complete clearing was attained, as confirmed by the visibility of the grids through the tissues, which enabled tissue imaging thereafter.”

From this description, it appears that the main limitation of the commercial kit lies in the refractive index (RI) matching medium, rather than the overall protocol. To help clarify this point and also to strengthen the evidence that PT-CLEAR3D performs better than the commercial kit, the manuscript would benefit from: representative LSFM images from samples processed with both PT-CLEAR3D and the commercial kit; quantitative assessment of signal-to-noise (S/N) and signal-to-background (S/B) ratios in both cases and a brief discussion of labeling specificity in both cases.

2.Since the tissues analyzed were unusually thick (from 0.5 to 50 mm³), signal attenuation and degradation at depth may significantly affect imaging quality. I recommend the authors provide representative images acquired at multiple z-positions (e.g., 50 µm, 100 µm, 200 µm, etc.) for human, marmoset, and macaque testes samples. This would demonstrate labeling uniformity throughout the depth of the section. Please also comment explicitly on whether signal intensity decreases with depth and indicate any correction methods used (e.g., deconvolution, intensity normalization).

3.For all figures with fluorescent images, please specify: the thickness of the imaged samples and whether the displayed images are maximum intensity projections (MIPs) or single optical sections. This information is essential for readers to correctly interpret resolution and signal fidelity.

4.The manuscript mentions the use of Arivis Pro TileSorter for manual stitching. Please provide more details such as the percentage of image overlaps used and how registration errors were corrected. This will improve reproducibility.

5.Please provide 1–2 sentences describing sample mounting/orientation during LSFM imaging (cuvette? Chamber? Suspended?) and explicitly state the immersion medium used. Confirm that the objectives were corrected for the BABB refractive index.

Reviewer #2: Traditional 3D tissue analysis using fluorescence microscopy relies on aligning serial sections, which is computationally intensive and prone to artefacts, or imaging of whole mounts with limited imaging depth. Optical clearing combined with light sheet fluorescence microscopy (LSFM) enables imaging of intact tissues while preserving 3D architecture. Although optical clearing has been optimised for many tissues, primate testis tissue remains a challenge due to its dense cellular structure.

This study presents an optimised clearing method for primate testis, PT-CLEAR3D, benchmarked against a commercial clearing kit and traditional confocal microscopy of sliced tissues. The approach enables more detailed structural analysis of testis tissue and will facilitate significant advancements for the field.

I believe this paper is worthy of publication, but I offer the following suggestions to strengthen it:

1. PT-CLEAR3D was tested only against one commercial solvent-based kit, not against widely used methods such as DISCO, or other BABB protocols. This limitation should either be noted, or additional comparisons included.

2. Line 366 – PTMCs are not introduced; please define.

3. Line 404 – Clarify that the commercial kit was inefficient specifically for testis tissue, not in general.

4. Figures 1A, 1E, 3A – Fluorescence LUTs should be more distinct; magenta is visually too close to red and should be replaced with e.g. green.

5. Figure S1D – Only 5 cells appear present in the group of 8 spermatagonia cells; label individual cells within each group e.g. with asterisks.

6. Figure 7 –

a. Ensure 2D and 3D images show equivalent fields of view. At present, the 2D images have a smaller field of view compared to the 3D images.

b. State whether local maximum projection was used for 3D rendering, and over what range. Ensure any maximum projection processing is consistent between 2D and 3D images.

**Do you want your identity to be public for this peer review?** For information about this choice, including consent withdrawal, please see our Privacy Policy

Reviewer #1: No

Reviewer #2: **Yes: ** Christopher Smith

---

## [Author Response · Author response to Decision Letter 1]

24 Sep 2025

To Su-Ren Chen

Academic Editor PLOS ONE

RE: Response Letter

PONE-D-25-22183

Title: Optical tissue clearing and 3D imaging of intact primate testicular tissue: a novel technology development

Dear Mrs. Chen.

Following positive evaluation by both reviewers and your invitation, we are delighted to submit our revised manuscript to PLOS ONE. We have addressed all points raised by the reviewers. We have generated new data, modified Figures 1, 3, 4, 6, 7 and S1 and generated new Figs. S2-S9.

Financial disclosure

The Deutsche Forschungsgemeinschaft (DFG) provided funding for staff and consumables (Clinical Research Unit #326) and for instrumentation:

- Zeiss LSM 980: INST 211/898-1 FUGB (Confocal Laser Scanning Microscope)

- Miltenyi UltraMicroscope II Super Plan: INST 211/899-1 FUGB (Light Sheet Microscope)

Data availabilitty

All authors agree that the data are publicly shared. We have already deposited the imaging data in BioImage Archive (https://doi.org/10.6019/S-BIAD1898) and made it publicly available. We consider the entire manuscript to be a protocol paper with all details provided in the material and methods and results sections.

Response to Reviewers

Reviewer #1: The study introduces an innovative method (PT-CLEAR3D) for tissue clearing and 3D imaging, enabling detailed analysis of spatial organization and cellular interactions within intact primate testes. This approach marks a significant advancement in the investigation of testicular physiology and pathophysiology, overcoming key limitations of conventional histology.

The manuscript is generally well-structured and clearly written, and the methodology is described in detail. Nevertheless, the manuscript would benefit from additional clarification.

Response: We appreciate the generally positive evaluation of our manuscript by this referee.

1.The authors state (line 406): “However, after subsequently incubating the same ITTs in BABB, complete clearing was attained, as confirmed by the visibility of the grids through the tissues, which enabled tissue imaging thereafter.”

From this description, it appears that the main limitation of the commercial kit lies in the refractive index (RI) matching medium, rather than the overall protocol. To help clarify this point and also to strengthen the evidence that PT-CLEAR3D performs better than the commercial kit, the manuscript would benefit from: representative LSFM images from samples processed with both PT-CLEAR3D and the commercial kit; quantitative assessment of signal-to-noise (S/N) and signal-to-background (S/B) ratios in both cases and a brief discussion of labeling specificity in both cases.

Response: We have performed additional work. We now show representative LSFM images from samples processed with the commercial kit (commercial clearing protocol, CCP) and from samples subjected to CCP followed by BABB clearing (S2 – S5 Figures). On these images, we performed signal-to-noise and signal-to-background measurements. Please note that the images used for comparison were not necessarily acquired in the exact same orientation or with identical acquisition, settings (for example laser power). Furthermore, in samples subjected to double processing with CCP and BABB, a decrease in fluorescence intensity was observed. This reduction is likely due to fluorophore degradation and quenching effects caused by prolonged chemical exposure, as well as the time elapsed between the first and second imaging sessions (approximately one week).

With regard to labeling specificity, staining was generally more specific in human tissue, which showed higher signal intensity and lower background levels. In non-human primate tissue, signal intensity was overall lower and background higher; however, labeling appeared consistent across depth.

During revision we have added a paragraph to material and methods and an additional reference on Image analysis Line 317 - 324. It reads:

Fluorescence signal was quantified across tissue depth using Fiji version 1.54 (Schindelin et al., 2012). At each depth (0–1500 µm, 250 µm steps), 20 regions were manually selected at sites of α-SMA–Cy3 staining outlining the testicular tubules. Square ROIs (10 × 10 pixels) were generated around each point, and corresponding background ROIs were obtained by translating them into the tubule interior (unstained region). Mean intensity and standard deviation were measured for each ROI. Signal-to-noise ratio (SNR) was calculated as SNR= (I_signal -I_background)/σ_background and signal-to-background ratio (SBR) as SBR = (I_signal )/I_background . YZ reslices were created along a horizontal line using Fiji.

Schindelin J, Arganda-Carreras I, Frise E, Kaynig V, Longair M, Pietzsch T, Preibisch S, Rueden C, Saalfeld S, Schmid B, Tinevez JY, White DJ, Hartenstein V, Eliceiri K, Tomancak P, Cardona A. Fiji: an open-source platform for biological-image analysis. Nat Methods. 2012 Jun 28;9(7):676-82. doi: 10.1038/nmeth.2019. PMID: 22743772; PMCID: PMC3855844.

Refer to supplementary figure and SNR/SBR measurements and labeling specificity in main text (lines 472 – 476).

2.Since the tissues analyzed were unusually thick (from 0.5 to 50 mm³), signal attenuation and degradation at depth may significantly affect imaging quality. I recommend the authors provide representative images acquired at multiple z-positions (e.g., 50 µm, 100 µm, 200 µm, etc.) for human, marmoset, and macaque testes samples. This would demonstrate labeling uniformity throughout the depth of the section. Please also comment explicitly on whether signal intensity decreases with depth and indicate any correction methods used (e.g., deconvolution, intensity normalization).

Response: We have added representative images (S6-S9) at multiple z-positions (250 µm - 2000 µm depth) for human, marmoset and macaque testis samples, with histogram brightness and contrast kept constant across all channels to enable a proper depth comparison.

In addition, we performed signal-to-noise ratio (SNR) and signal-to-background ratio (SBR) measurements. These analyses show limited signal intensity decrease in human samples (while starting at a generally higher level), but no appreciable decrease in marmoset and macaque tissues. This suggests that, despite the unusually large tissue volumes analyzed (0.5–50 mm³), labeling uniformity and imaging depth are well preserved using PT-CLEAR3D.

Finally, we confirm that no post-acquisition correction methods (e.g., deconvolution, intensity normalization) were applied, as no quantitative image analysis or automated segmentation was performed.

Refer to supplementary figure and SNR/SBR measurements and labeling specificity in main text (lines 476 - 479).

3.For all figures with fluorescent images, please specify: the thickness of the imaged samples and whether the displayed images are maximum intensity projections (MIPs) or single optical sections. This information is essential for readers to correctly interpret resolution and signal fidelity.

Response: We have added tissue volumes for the representative images showing the relevant dimensions width (X) x height (Y) x depth (Z) respectively. Tissue volumes as depicted by the measure box as visualized in 3D are shown in a modified Supplementary Figure 1. Please note that only the region of interest (ROI) is shown in these images and therefore, they do not represent the entire optically cleared tissue volumes. Human testicular tissue measuring 3 mm x 3 mm x 4 mm totaling 36 mm3, was used to generate Figs 5A, 6A, 7A, 7D, S1 and S3 movies and S1_FigA. From the same patient, testicular tissue with a thickness of 0.6 mm3 measuring 1 mm x 1 mm x 0.6 mm was used to generate Figs 6B-D, and S4 movie. From a different human testicular tissue with a volume of 1 mm3 and measuring 1 mm x 1 mm x 1 mm, Figs 4A and 5B and S2 movie were generated. The marmoset tissue had a volume of 29 mm3 with 3.2 mm x 3.2 mm x 2.8 mm dimensions. Lastly, the macaque tissue dimensions were 3.2 mm x 3.2 mm x 2.4 mm totaling 25 mm3. The images shown in 3D, are either maximum intensity or volumetric projections of the tissues while single optical planes were used for the 2D, therefore, no projections were used.

We have modified the figure legends accordingly and added a few details to materials and methods (lines 210-218).

4.The manuscript mentions the use of Arivis Pro TileSorter for manual stitching. Please provide more details such as the percentage of image overlaps used and how registration errors were corrected. This will improve reproducibility.

Response: We have revised the description in material and methods (lines 286-291) in accordance to the reviewers comments. Mosaic image acquisition of large samples was performed by generating multiple tiled stacks with a 20% overlap, which were later stitched to reconstruct full volumes. Tiled images were pre-aligned in Zeiss Arivis Pro TileSorter using grid mode by specifying the number of columns and rows, the acquisition order (“Straight, Rows”), and the image overlap (20%). To ensure accurate registration, the alignment was manually adjusted based on prominent features, such as αSMA-positive blood vessels, throughout the volume. Once the alignment was satisfactory, the tiles were stitched to generate the final mosaic images.

5.Please provide 1–2 sentences describing sample mounting/orientation during LSFM imaging (cuvette? Chamber? Suspended?) and explicitly state the immersion medium used. Confirm that the objectives were corrected for the BABB refractive index.

Response: As requested by the referee, we extended the description in material and methods. Samples were mounted in a cuvette filled with BABB solution and oriented perpendicular to the light sheet path for LSFM imaging. Emission was collected using 4× (NA 0.35, MI PLAN, DC57 WD16 O1 dipping cap) and 12× (NA 0.53, MI PLAN, DC57 WD10 O1 dipping cap) objectives, both corrected for the BABB refractive index (n = 1.56). Lines 268 -271.

Reviewer #2: Traditional 3D tissue analysis using fluorescence microscopy relies on aligning serial sections, which is computationally intensive and prone to artefacts, or imaging of whole mounts with limited imaging depth. Optical clearing combined with light sheet fluorescence microscopy (LSFM) enables imaging of intact tissues while preserving 3D architecture. Although optical clearing has been optimised for many tissues, primate testis tissue remains a challenge due to its dense cellular structure.

This study presents an optimised clearing method for primate testis, PT-CLEAR3D, benchmarked against a commercial clearing kit and traditional confocal microscopy of sliced tissues. The approach enables more detailed structural analysis of testis tissue and will facilitate significant advancements for the field.

I believe this paper is worthy of publication, but I offer the following suggestions to strengthen it:

Response: We are thankful for a positive evaluation of our study.

1. PT-CLEAR3D was tested only against one commercial solvent-based kit, not against widely used methods such as DISCO, or other BABB protocols. This limitation should either be noted, or additional comparisons included.

Response: We agree with the reviewer and noted the limitation as follows: “Noteworthy, PT-CLEAR3D that includes BABB as the clearing step was tested against only one commercial kit using the primate testes. Its compatibility with other clearing protocols and in other organs was not tested in this work.” The text was added to the discussion. Line 700 – 702.

2. Line 366 – PTMCs are not introduced; please define. Response: Done as requested (see line 430 – 431).

3. Line 404 – Clarify that the commercial kit was inefficient specifically for testis tissue, not in general.

Response: In response to the reviewer, the following sentence was added in lines 479 - 480: The efficiency of the commercial immunolabelling and clearing kit was tested with the primate testis only and not with other organs. We cannot judge its usefulness for other applications.

4. Figures 1A, 1E, 3A – Fluorescence LUTs should be more distinct; magenta is visually too close to red and should be replaced with e.g. green. Response: Done as requested. Figures were revised.

5. Figure S1D – Only 5 cells appear present in the group of 8 spermatagonia cells; label individual cells within each group e.g. with asterisks.

Response: We have added Fig S1D1 and revised the figure legend as follows: Spermatogonia cells in each group have been circled using different colors due to difficulties labelling each cell with asterisks. For this, single cells are circled in orange, pairs in cyan, groups of four in green circle while a group of eight cells is not shown. However, in a group of 6 cells circled in white (inset) four individual cells are marked with white arrows and green arrows partially shows two cells whose adjoining parts were sliced off during collection of the tissue.

6. Figure 7 –

a. Ensure 2D and 3D images show equivalent fields of view. At present, the 2D images have a smaller field of view compared to the 3D images.

b. State whether local maximum projection was used for 3D rendering, and over what range. Ensure any maximum projection processing is consistent between 2D and 3D images.

Response: Done as requested. Please note: The 2D images are single optical sections and thus, there was no projection of data whereas the 3D images are image projections generated from the data sets. Therefore, the fields of view are different in all species - human, marmoset and macaque. However, the 2D image positions are marked in the 3D images using a rectangle for perspective while the projection ranges for the regions of interest are presented by the measure boxes in a modified Supplementary Figure 1. Maximum projections were rendered in 3D and subsequently, volumetric and maximum intensity were used for image analysis. Descriptions of the different views were added to Lines 605 – 608.

---

## [Decision Letter · Decision Letter 1]

12 Oct 2025

Dear Dr. Schlatt,

Thank you for submitting your manuscript to PLOS ONE. After careful consideration, we feel that it has merit but does not fully meet PLOS ONE’s publication criteria as it currently stands. Therefore, we invite you to submit a revised version of the manuscript that addresses the points raised during the review process.

We look forward to receiving your revised manuscript.

Kind regards,

Su-Ren Chen

Academic Editor

PLOS ONE

Journal Requirements:

Additional Editor Comments:

Reviewer 2 raises some additional minor concerns. I have to ask the authors to further revise the manuscript according to his/her suggestions. I am looking forwards to receive your final version.

Reviewers' comments:

Reviewer's Responses to Questions

**Comments to the Author**

Reviewer #1: All comments have been addressed

Reviewer #2: All comments have been addressed

2. Is the manuscript technically sound, and do the data support the conclusions?

Reviewer #1: Yes

Reviewer #2: Partly

3. Has the statistical analysis been performed appropriately and rigorously?

Reviewer #1: Yes

Reviewer #2: No

4. Have the authors made all data underlying the findings in their manuscript fully available?

Reviewer #1: Yes

Reviewer #2: Yes

5. Is the manuscript presented in an intelligible fashion and written in standard English?

Reviewer #1: Yes

Reviewer #2: Yes

Reviewer #1: The authors have adequately addressed all the comments; therefore, no further revisions are needed on my part

Reviewer #2: Thank you to the authors for thoughtfully addressing each of my previous comments. I have a couple of additional recommendations:

• Lines 466–482: The purpose of this paragraph is to compare PT-CLEAR3D with the other clearing techniques and to justify why PT-CLEAR3D represents the optimal approach. However, with the inclusion of new information regarding the analysis of the different clearing techniques, as well as the clarification that the testing was conducted only on testis tissue and not on other tissue types, the concluding sentence of the paragraph now appears somewhat disjointed. I recommend revising the structure of this final paragraph to improve its coherence and logical flow.

• I commend the authors for their additions in response to the first reviewer, which in my view clearly demonstrate the benefit of additional BAAB clearing following the use of the commercial clearing kit in primate testis tissue (Figs. S2–S5). However, it is unexpected that PT-CLEAR3D was omitted from this analysis. Including PT-CLEAR3D in these comparative analyses would substantially strengthen the overall message of the manuscript.

• There appears to be a large standard deviation in Figs. S5 and S9. The authors should provide statistical evidence to indicate whether there are significant differences between the clearing techniques at depth.

**Do you want your identity to be public for this peer review?** For information about this choice, including consent withdrawal, please see our Privacy Policy

Reviewer #1: No

Reviewer #2: **Yes: ** Christopher Smith

---

## [Author Response · Author response to Decision Letter 2]

5 Nov 2025

To Su-Ren Chen

Academic Editor PLOS ONE

RE: Response Letter

PONE-D-25-22183

Title: Optical tissue clearing and 3D imaging of intact primate testicular tissue: a novel technology development

Dear Mrs. Chen.

We appreciate the responses of both reviewers and are delighted that reviewer 1 accepted our revision. We have dealt with the comments of reviewer 2 during a second round of revision and resubmit the revised manuscript. You find our response to the three comments and appropriate changes in the text.

Response to Reviewers

Reviewer #1: The authors have adequately addressed all the comments; therefore, no further revisions are needed on my part

Reviewer #2: Thank you to the authors for thoughtfully addressing each of my previous comments. I have a couple of additional recommendations:

Comments by reviewer 2

1) Lines 466–482: The purpose of this paragraph is to compare PT-CLEAR3D with the other clearing techniques and to justify why PT-CLEAR3D represents the optimal approach. However, with the inclusion of new information regarding the analysis of the different clearing techniques, as well as the clarification that the testing was conducted only on testis tissue and not on other tissue types, the concluding sentence of the paragraph now appears somewhat disjointed. I recommend revising the structure of this final paragraph to improve its coherence and logical flow.

Answer: We have rewritten the summarizing paragraph to increase the coherence (Lines 455-476).

2) I commend the authors for their additions in response to the first reviewer, which in my view clearly demonstrate the benefit of additional BAAB clearing following the use of the commercial clearing kit in primate testis tissue (Figs. S2–S5). However, it is unexpected that PT-CLEAR3D was omitted from this analysis. Including PT-CLEAR3D in these comparative analyses would substantially strengthen the overall message of the manuscript.

Answer: We appreciate the reviewer’s commendation for demonstrating BABB’s beneficial effect on rescuing clearing efficiency following the use of the commercial kit (Figs. S2–S5).

We are showing the results of our PT-CLEAR3D in Fig. S6-S9. The additional figures S2-S9 were added on request by the first reviewer to address signal to background ratio and depth. This has determined the order of the figures added during revision 1.

To address the reviewer’s suggestion, we have now expanded the manuscript to provide detailed descriptions of the comparative analyses in the main text (Lines 443-454), which are already visually represented in figures illustrating depth-dependent image quality (Figs. S2–S9). We assessed signal-to-noise ratio (SNR) and signal-to-background ratio (SBR) across all clearing conditions – CCP, CCP + BABB, and PT-CLEAR3D – in corresponding figures. While we elected not to aggregate all measurements into a single figure due to substantial differences in imaging conditions (such as laser power, one-sided light sheets vs two-sided, time of acquisition) between Figs. S2–S5 and Figs. S6–S9, we have revised the manuscript to detail results for all conditions. We acknowledge that this introduces potential limitations in direct SNR/SBR comparison due to variations in experimental parameters. Nevertheless, we believe the reported trends – including improved SNR/SBR in human tissues accompanied by slight depth-dependent decreases, and relatively lower SNR/SBR in macaque and marmoset tissues potentially attributable to less efficient antibody labelling without significant depth-dependent degradation – remain consistent across all conditions.

3) There appears to be a large standard deviation in Figs. S5 and S9. The authors should provide statistical evidence to indicate whether there are significant differences between the clearing techniques at depth.

Answer: We thank the reviewer for this observation. The large standard deviations in Figures S5 and S9 reflect the intrinsic variability of the measurements rather than replicate-based variation. These measurements were performed on representative samples from both clearing protocols.

Signal intensity was quantified at 20 randomly selected positions per optical plane at different depths throughout the tissue. However, testicular tissue is inherently heterogeneous, and local variations in cell density and refractive index can influence light propagation. Given the effective tissue thickness shown, each measurement point may differ, leading to variable fluorescence attenuation. Additionally, regions closer to the periphery of the light sheet generally receive stronger illumination. At the same time, attenuation occurs not only along the z-axis (depth) but also laterally (y-axis) due to imperfect clearing.

Consequently, a statistical comparison between clearing methods based on single-sample datasets does not provide a meaningful or reliable assessment of performance. Instead, qualitative evaluation of the raw LSFM images (Fig. S2-S4, S6-S8) more clearly illustrates the differences in optical transparency and image quality between the protocols—particularly the presence of blur and reduced resolution in less efficiently cleared samples, which may not be fully captured by simple intensity-based metrics.

---

## [Editor Report · Decision Letter 2]

9 Nov 2025

Optical tissue clearing and 3D imaging of intact primate testicular tissue: a novel technology development

PONE-D-25-22183R2

Dear Dr. Schlatt,

We’re pleased to inform you that your manuscript has been judged scientifically suitable for publication and will be formally accepted for publication once it meets all outstanding technical requirements.

Kind regards,

Su-Ren Chen

Academic Editor

PLOS ONE

Additional Editor Comments (optional):

I would like to recommend acceptance of your revised manuscript (R2).
---

## [Editor Report · Acceptance letter]

PONE-D-25-22183R2

PLOS ONE

Dear Dr. Schlatt,

I'm pleased to inform you that your manuscript has been deemed suitable for publication in PLOS ONE. Congratulations! Your manuscript is now being handed over to our production team.

Kind regards,

on behalf of

Prof. Su-Ren Chen

Academic Editor

PLOS ONE